# Genome-Wide Pathway Exploration of the *Epidermidibacterium keratini* EPI-7^T^

**DOI:** 10.3390/microorganisms11040870

**Published:** 2023-03-28

**Authors:** Yunseok Oh, Seyoung Mun, Young-Bong Choi, HyungWoo Jo, Dong-Geol Lee, Kyudong Han

**Affiliations:** 1Department of Bioconvergence Engineering, Dankook University, Jukjeon, Yongin 16890, Republic of Korea; 72210364@dankook.ac.kr; 2Department of Microbiology, College of Science & Technology, Dankook University, Cheonan 31116, Republic of Korea; 12200281@dankook.ac.kr (S.M.); chohw@cosmax.com (H.J.); 3Center for Bio Medical Engineering Core Facility, Dankook University, Cheonan 31116, Republic of Korea; 4Department of Chemistry, College of Science & Technology, Dankook University, Cheonan 31116, Republic of Korea; chem0404@dankook.ac.kr; 5R&I Center, COSMAX BTI, Pangyo-ro 255, Bundang-gu, Seongnam 13486, Republic of Korea; 6R&D Center, HuNBiome Co., Ltd., Gasan Digital 1-ro, Geumcheon-gu, Seoul 08507, Republic of Korea

**Keywords:** *Epidermidibacterium keratini* EPI-7^T^, skin anti-aging, de novo whole-genome sequencing, comparative genomic analysis, anti-aging, biosynthetic gene, 20 amino acids, orotic acid, functional cosmetic industry

## Abstract

Functional cosmetics industries using skin microbiome screening and beneficial materials isolated from key microorganisms are receiving increasing attention. Since *Epidermidibacterium keratini* EPI-7^T^ was first discovered in human skin, previous studies have confirmed that it can produce a new pyrimidine compound, 1,1′-biuracil, having anti-aging effects on human skin. Therefore, we conducted genomic analyses to judge the use value of *E. keratini* EPI-7^T^ and provide up-to-date information. Whole-genome sequencing analysis of *E. keratini* EPI-7^T^ was performed to generate new complete genome and annotation information. *E. keratini* EPI-7^T^ genome was subjected to comparative genomic analysis with a group of closely-related strains and skin flora strains through bioinformatic analysis. Furthermore, based on annotation information, we explored metabolic pathways for valuable substances that can be used in functional cosmetics. In this study, the whole-genome sequencing (WGS) and annotation results of *E. keratini* EPI-7^T^ were improved, and through comparative analysis, it was confirmed that the *E. keratini* EPI-7^T^ has more metabolite-related genes than comparison strains. In addition, we annotated the vital genes for biosynthesis of 20 amino acids, orotic acid, riboflavin (B2) and chorismate. In particular, we were able to prospect that orotic acid could accumulate inside *E. keratini* EPI-7^T^ under uracil-enriched conditions. Therefore, through a genomics approach, this study aims to provide genetic information for the hidden potential of *E. keratini* EPI-7^T^ and the strain development and biotechnology utilization to be conducted in further studies.

## 1. Introduction

The *Epidermidibacterium keratini* EPI-7^T^ was first introduced by Lee DG et al. in 2018. In a previous study, a novel bacterial strain, EPI-7^T^, was isolated from skin samples (human epidermal keratinocytes) and characterized by taxonomic identification [1]. In addition, it has been proven that *E. keratini* EPI-7^T^ culture solutions have a positive effect on the improvement of anti-aging-related gene expression in Hs68 cells to UV-irradiated fibroblast cells [2]. Moreover, a recent study explored anti-aging compounds produced by *E. keratini* EPI-7^T^ (orotic acid, 1,1′-biuracil) depending on the precursor added to the culture medium [3]. Through previous studies, *E. keratini* EPI-7^T^ is regarded as a hidden beneficial microbe with potential applicabilities in skin anti-aging, skincare, cosmetics, the human beauty industry, and the skin microbiome [1,2,3]. However, fundamental genomics analysis and genomic resource research based on the *E. keratini* EPI-7^T^ genome has not progressed much since 2021. The lack of genomic resource research makes it difficult to understand the organism’s potential molecular and biological processes. Furthermore, the closely-related microorganisms are still not defined, and thus the study of the microbial genome and classification of functional genes is important in ensuring bacterial secondary metabolites and socioeconomic benefits that are not yet known in the *E. keratini* EPI-7^T^.

With the advancement of next-generation sequencing (NGS) technology and the reduction in sequencing costs, many microbial genome data have been published and used for further examination, such as genetic screening, functional characterization, metabolism pathway prediction, and resource search [4,5]. NGS technology has lowered barriers to entry for the genetic approach at the genome level and allowed deeper evolutionary analysis. In this study, bacterial whole-genome sequencing (WGS) has been performed using Illumina NovaSeq 6000 platform to generate paired-end (PE) libraries (short-read). In addition, long-read sequencing data generated by the SMRTbell library and PacBio RSII platform is also often used in the construction of a bacterial genome [6,7]. These PE and long-read sequencing data have been involved in hybrid de novo assembly, depending on the research purposes. Here, we also choose the de novo whole-genome assembly strategy for the construction of the precise *E. keratini* EPI-7^T^ genome.

Recently, broadly-used and systematic web-based tools that can perform a comparative analysis of strains along with increasing WGS data of bacteria are continuously developed [8,9]. PATRIC enables an accurate bioinformatic analysis using publicly and quickly available microbial genome sequences by supporting eight services, such as web-based sophisticated comparative analysis and comprehensive genome analysis [10,11]. In addition, OrthoVenn2 efficiently performs functional and evolutionary analysis of proteins across multiple species, comparative genome studies, and visualization of taxonomic evidence, through genome-wide comparative analysis of orthologous clusters with free access to a user-friendly web server [12]. EggNOG-mapper also provides several functional annotation sources, such as the KEGG pathway, Gene Ontology labels, EC numbers, and COG functional categories [13]. In addition, the large metagenomic data sets in EggNOG software make possible faster annotation than Prokka, which is considered to be the fastest annotation tool for annotating prokaryotes [14]. Therefore, we performed the pan-genome analysis pipeline (PGAP) using the seven most closely-related species with *E. keratini* EPI-7^T^ (*Sporichthya polymorpha*, *Modestobacter marinus*, *Modestobacter roseus*, *Modestobacter versicolor*, *Blastococcus aggregatus*, *Geodermatophilus obscurus*, and *Frankia alni*) and two species belonging to the class *acinomycetia* (*Streptomyces klenkii* and *Streptomyces griseocarneus*) to identify the genetic features that are fundamentally essential and specific to *E. keratini* EPI-7^T^. To establish the dermatophilic status of EPI-7 in the microbiome, we initially investigated the microbial taxonomy between *E. keratini* EPI-7^T^ and 20 species of the prominent skin microbiota in human using the UBCG (up-to-date bacterial core gene) pipeline and thus selected 7 major species [15,16,17]. Additionally, PATRIC, OrthoVenn2, and EggNOG-mapper were performed with *E. keratini* EPI-7^T^ and seven major species (*Cutibacterium acnes*, *Staphylococcus epidermidis*, *Staphylococcus aureus*, *Staphylococcus warneri*, *Micrococcus luteus*, *Streptococcus pyogenes*, and *Streptococcus mitis*) known as human skin flora [18,19]. We confirmed the genetic ability of skin-derived *E. keratini* EPI-7^T^ to have similarities with the skin flora species and its unique genetic abilities.

Bacterial cellulose, known to have biocompatible, hydrophilic, and biodegradable biopolymer properties, is used as an emulsion stabilizer and skin treatment in cosmetics [20]. In addition, biomolecules such as secondary metabolites of bacterial origin, enzymes, pigments, and exopolysaccharides are attracting attention for commercial and cosmetic applications [21,22]. Particularly, it was confirmed that the gene cluster involved in the orotic acid biosynthesis pathway mentioned in the previous study was present in the *E. keratini* EPI-7^T^ chromosomal loci. Furthermore, the genes involved in the biosynthesis pathway of 20 amino acids and metabolites that can affect the skin while inhabiting the skin were identified.

Hence, we perform a WGS and comparative genomic analysis of *E. keratini* EPI-7^T^. This study predicts the evolutionary relationship of *E. keratini* EPI-7^T^ and the effect of substrate added to the culture medium on the pathway and compound accumulations. By providing a more accurate and up-to-date complete genome following its basic information, we provide bioinformatical clues to further study *E. keratini* EPI-7^T^ and encourage sharper demonstrations and industrial applications on the development of biomaterials and cosmetic compounds.

## 2. Materials and Methods

### 2.1. Bacterial Materials and DNA Extraction

*E. keratini* EPI-7^T^ used for the WGS and hybrid de novo assembly was provided by COSMAX BTI R&I Center. *E. keratini* EPI-7^T^ was cultured on an R2A agar medium at optimal conditions (R2A agar at 25 °C, pH 6.0, absence of NaCl) for 3–5 days. Then, a single colony was inoculated in an R2A broth medium and cultured for 3–5 days under optimal conditions. After five days of incubation, stock and DNA extraction were executed. The stock was made so that the final glycerol was 25% and stored at −80 °C. This genomic DNA of *E. keratini* EPI-7^T^ from cultured in an R2A broth medium was extracted using the Qiagen DNeasy UltraClean Microbial Kit (Qiagen, Hilden, Germany), and all experimental processes were performed following the optimal protocols provided by the DNA extraction kit. The quality check of all extracted DNA was conducted using the NanoDrop One (ThermoFisher Scientific, Waltham, MA, USA) equipment.

### 2.2. Whole-Genome Sequencing and Hybrid De Novo Assembly

*E. keratini* EPI-7^T^ genomic DNA was sequenced at the Teragen Bio (Theragen Bio, Seongnam, Republic of Korea) by Illumina NovaSeq 6000 (Illumina Inc., San Diego, CA, USA). Illumina PE libraries were prepared using the TruSeq Nano DNA Prep kit (Illumina, San Diego, CA, USA) in terms of the included instructions. Subsequently, we prepared the PacBio SMRTbell raw sequence data downloaded from COSMAX R&I Center, Seongnam, Republic of Korea.

With the Illumina PE libraries and PacBio SMRTbell raw sequence data, the *E. keratini* EPI-7^T^ genome was assembled with the de novo hybrid strategy. First, the Canu v1.7 generated long-read assembly contigs from PacBio SMRTbell raw sequence data with default option [23]. Next, to correct and improve accuracy, Illumina PE reads were integrated onto the draft genome using BWA mem v.0.7.17 (Wellcome Trust Sanger Institute, Cambridge, UK) [24]. Then, a polishing round was performed five times using pilon v.1.22 (Broad Institute of MIT and Harvard, Cambridge, MA, USA) [25].

### 2.3. Genomic Gene Prediction and Genome Annotation

The newly-assembled genome was used for gene prediction and annotation using Rapid Prokaryotic Genome Annotation (Prokka) ver1.14.6 with the RNAmmer and Addgenes [26,27]. Structural annotation was conducted by using Prodigal, Aragorn, Barrnap, and MinCED were used to predict coding sequence (CDS), tRNA, rRNA, and CRISPRs assay, respectively [28,29,30,31]. For functional annotation, genes were searched against the UniProt/Swissprot and NCBI non-redundant proteins RefSeq databases using BLASTP v2.2.29+ with an E-cutoff value of 1.0 × 10^−10^ [32,33,34]. Then, protein domains were also searched against the Pfam databases using InterProScan v.5.19-58.0 [35,36].

### 2.4. Improvement of Whole-Genome Sequencing and Gene Annotation for E. keratini EPI-7^T^

For comparison between the newly-assembled genome and the previous genome in NCBI, the direction of the two genomes was aligned based on the newly-assembled genome using Chromeister (https://usegalaxy.org/ (accessed on 12 September 2021)), PATRIC 3.6.12 (University of Málaga, Málaga, Spain) with genome alignment (Mauve) services [11,37]. Moreover, the difference between the two assembled genomes was compared using BioEdit 7.2 (North Carolina State Unicersity, Raleigh, NC, USA) [38]. In addition, gene annotation was improved by excluding hypothetical protein and domain of the unknown function (DUF) through manual inspection between the newly-annotated data and the previous one registered in NCBI.

### 2.5. Analysis of Pan-Genome and Comparative Whole-Genome Analysis

With the hybrid de novo assembled genome for pan-genome and comparative analysis, the comparable bacterial genome data of nine comparison strains were downloaded from the NCBI genome datasets (https://www.ncbi.nlm.nih.gov/datasets/ (accessed on 14 October 2021)) (Appendix A). Furthermore, pan-genome analysis was conducted with a group of the given strains closely related to *E. keratini* EPI-7^T^. The pan-genome analysis pipeline (PGAP) 1.2.1 was used for pan-genome analysis [39]. PGAP has five analysis modules: (i) cluster analysis of functional genes, (ii) pan-genome profile analysis, (iii) genetic variation analysis, (iv) species evolution analysis, and (v) function enrichment analysis. The multi-paranoid (MP) method was used for cluster analysis of functional genes and showed the orthologous genes presented in the comparison strains. Moreover, the phylogenetic analysis based on the core genome was performed in the MEGA 11 program with 1000 bootstrap replications for checking an evolutionary distance [40]. Phylogenetic analysis was conducted using the neighbor-joining method with the maximum composite likelihood model.

Genome-based phylogeny analysis between *E. keratini* EPI-7^T^ and 20 human skin microbiota (*Cutibacterium granulosum*, *Cutibacterium avidum*, *Cutibacterium acnes*, *Staphylococcus aureus*, *Staphylococcus hominis*, *Staphylococcus haemolyticus, Staphylococcus lugdunensis*, *Staphylococcus warneri*, *Staphylococcus epidermidis, Micrococcus luteus*, *Streptococcus pyogenes*, *Streptococcus agalactiae*, *Streptococcus mitis*, *Corynebacterium jeikeium*, *Corynebacterium aurimucosum*, *Pseudomonas aeruginosa*, *Acinetobacter pittii*, *Acinetobacter baumannii*, *Roseomonas mucosa*, *Neisseria meningitidis*) was performed using UBCG pipeline v3.0 (https://www.ezbiocloud.net/tools/ubcg, (accessed on 15 October 2021)), which is based on 92 core genes that exist as a single copy [17,41,42] (Appendix A). A phylogenetic tree was visualized using the MEGA 11 program (Toky Metropolitan University, Tokyo, Japan) [40]. Comparative whole-genome analysis was also performed with a group of seven major species (Cutibacterium acnes, Staphylococcus epidermidis, Staphylococcus aureus, Staphylococcus warneri, Micrococcus luteus, Streptococcus pyogenes, and Streptococcus mitis) known as skin flora (Appendix A). Web-based genome comparison tools, such as PATRIC (https://www.patricbrc.org (accessed on 15 October 2021)), OrthoVenn2 (https://orthovenn2.bioinfotoolkits.net (accessed on 15 October 2021)) and EggNOG-mapper (http://eggnog-mapper.embl.de (accessed on 14 March 2022)) were used [11,12,13]. PATRIC confirmed the amino acid (AA) identity between *E. keratini* EPI-7^T^ and the comparison strains using proteome analysis services with the default setting. OrthoVenn2 also confirmed similarity and common gene clusters by comparing orthologous genes using the AA sequence between *E. keratini* EPI-7^T^ and the comparison strain; E-value: 1.0 × 10^−2^, inflation value: 1.5. The EggNOG-mapper was used for functional annotation and orthology assignments by submitting the amino acid sequence data; minimum hit e-value: 0.001, minimum hit bit-score: 60, percentage identity: 40, minimum % of query coverage: 20, minimum % of subject coverage: 20.

### 2.6. Prediction of the Particular Biosynthesis Pathway in E. keratini EPI-7^T^

Functional analysis was performed through manual inspection based on the annotation data in this study. Based on Appendix A, the annotation-based manual inspection resulted in identification of biosynthetic pathways of amino acids or compounds in bacteria and necessary gene discovery concerning MetaCyc (MetaCyc.org), KEGG pathway (http://www.genome.jp/kegg/ (accessed on 11 October 2021)), and other studies [21,43,44,45].

## 3. Results

### 3.1. Hybrid De Novo Assembly and Genomic Characteristics of E. keratini EPI-7^T^

The overall experiment was performed in a serial process (Figure 1). Before the NGS data production, the result of the 16S rRNA gene full-length Sanger sequencing of prepared bacterial gDNA matched 99.50% identity to the *Epidermidibacterium keratini* strain EPI-7^T^ chromosome (Appendix A). An Illumina PE libraries generated 14,057,360 reads covering 2,122,661,360 bp (2.12 Gb) of sequence data, the Q30 (more base rate, nucleotide accuracy >99.9% rate) rate was 92.13%, and the GC content was 67.70%. PacBio SMRTbell sequence data comprised 75,720 reads covering 680,622,771 bp (0.68 Gb), and mean length was 8989 bp, N50 was 12,811, and average GC content was 65.17% (Appendix A). As a result, a single chromosome (a contig) of *E. keratini* EPI-7^T^ was obtained through hybrid de novo assembly and polishing using Canu, BWA mem, and Pilon. The total *E. keratini* EPI-7^T^ genome length was 4,018,778 bp, and the average GC content was 67.34% (67.34%) (Table 1).

### 3.2. Genomic Gene Prediction and Annotation

The *E. keratini* EPI-7^T^ genome had 3851 predicted genes (3799 CDSs, 3 rRNAs, 48 tRNAs, and 1 tmRNA) and 1 CRISPR array (Appendix A). Of the 3851 predicted genes of *E. keratini* EPI-7^T^, except for 13 CDSs, 51 RNA genes, and 1 CRISPR array, 3786 CDSs had annotated in the UniProt, NCBI nr, InterProScan (respectively, 2754, 3779, 3509, accounting for 72.49%, 99.47%, 92.37% of the total number of annotated genes) (Appendix A). Among the 3786 annotated CDSs, 3169 genes (83.70%) were annotated functional genes, and 617 (113 + 504) genes (16.30%) were unknown (DUF) or hypothetical genes.

### 3.3. Comparison between Newly-Generated and Previous Genome and Annotation

The size of the genome in the NCBI used for comparison with the newly-assembled genome was 4,018,808 bp, which is 30 bp longer than the newly-assembled genome. Moreover, we identified significant differences in the comparison results between the two genomes, which were aligned with the sequence orientation using the genome alignment service (Mauve) in PATRIC. First, in the case of sequence orientation, when the newly-assembled genome was referenced, it was found that the sequences of strands with different senses were represented (Appendix A). Second, the alignment results of the two genomes show that the region of 1 bp to 2,090,211 bp of the newly-assembled genome complementarily matched 2,090,242 bp to 1 bp of the previous genome, and the remaining 2,090,212 bp to 4,018,778 bp region complementarily matches 4,018,808 bp to 2,090,243 bp (Figure 2). As a result, the difference between the two genome sequences is about 70 bp, which shows that the newly-assembled genome is 99.998258% similar. Five single nucleotide gaps and one small gap were filled. The newly-assembled genome corrected two single nucleotide gaps and one duplication error. We also found four SNV mismatches and corrected them in this study (Appendix A).

Using the corrected *E. keratini* EPI-7^T^ genome, we were able to identify a total of 3851 genes, compared to the previous genome data. Among a total of annotated genes, 3799 CDSs were implicated in the gene annotation process. As a result, the number of annotated genes increased from the previous 3753 (98.79%) to 3786 (99.66%). Through an accurate gene annotation and the better-assembled genome data, we finally mined 3169 functional genes, a higher number than the 3012 previous annotated genes. The annotated data produced in this study showed more information on 157 functional genes. The number of unknown (DUF) and hypothetical genes were 113 and 504, respectively. A total of 3 rRNA, 48 tRNA, 1 tmRNA, and 1 CRISPR cluster were searched out using Aragorn, Barrnap, and MinCED software (Table 2) [28,29,30].

### 3.4. Core and Pan-Genome Analysis with Closely-Related Species Group

Pan-genome analysis showed that the overall orthologous gene cluster of *E. keratini* EPI-7^T^ and the 9 comparison strains had 15,363 pan-genomes. A comparative pan-genome analysis was conducted to study genomic diversity and evolutionary relationships, and the comparison strains were chosen by considering taxonomy relations based on the 16s RNA gene. The pan-genome and core genome size, the number of accumulated genes about the number of genomes, could be predicted by Heaps’ law (y = A*x**B + C, where x is the number of genomes, y is the size of the pan-genome, * is the multiplication sign (×), ** is the square sign (^), and A, B, and C fit parameters) [46,47]. Therefore, the pan-genome size steadily increased as the comparison strains genomes were added, and it was confirmed that the pan-genome formula (y = 3924.78591896411 *x**0.589 + 71.877334572596) was 0 < B < 1 (Figure 3A). Correspondingly, the orthologous gene cluster shared by *E. keratini* EPI-7^T^ and all comparison strains had 1032 core genomes. Then, the core genome size was maintained after a drastic decrease as the comparison strains genomes were added, and the core genome formula was y = 7798.73246740606 *exp (−1.029 * x) + 1175.52917424416 (Figure 3A). However, the pan-genome of 15,363 orthologous gene clusters was divided into the core genome of 1032 orthologous gene clusters (6.72%) present in all 10 strains, the accessory genome of 5236 orthologous gene clusters (34.08%) partially present in 10 strains, and 9095 strain-specific orthologous gene clusters (59.20%) (Figure 3B). Furthermore, these results not only show that very few orthologous gene clusters were shared between *E. keratini* EPI-7^T^ and the comparison strains, but also suggest that *E. keratini* EPI-7^T^ has very different genetic repertories, such as metabolic ability, production of secondary metabolites, and host cell communications. A phylogenetic tree was constructed based on the core genome to confirm further the evolutionary relationship between *E. keratini* EPI-7^T^ and the comparison strain. The branch of *E. keratini* EPI-7^T^ shows a significant evolutionary distance from other comparative strains (Figure 3C). Thus, the pan-genome analysis indicates that *E. keratini* EPI-7^T^ branched independently from other strains in the phylogenetic tree constructed based on the core genome and that *E. keratini* EPI-7^T^ had low genetic similarity with the other strains.

### 3.5. Comparative Genomic Analysis with Skin Flora Group

To identify suitable comparison candidates, we conducted a taxonomic analysis of the *E*. *keratini* EPI-7^T^ strain and 20 species of the human skin microbiota using the UBCG pipeline. *Micrococcus luteus* has the closest genetic correlation to *E*. *keratini* EPI-7^T^ (Figure 4). Based on these results, an in-depth comparative genetic characterization and specificity analysis was performed between the seven major skin microbiota and *E*. *keratini* EPI-7^T^. The integrated and comparative analysis of genomic and gene associations was investigated using web-based PATRIC, OrthoVenn2, and EggNOG-mapper. Proteome comparison analysis based on amino acid sequence using PATRIC showed that *E. keratini* EPI-7^T^ had a low protein sequence identity with the skin flora group (Figure 5). The hit ratio to the amino acid sequence of *E. keratini* EPI-7^T^ was less than 50%, except for *Micrococcus luteus* (50.73%). In addition, most amino acid identity (AAI) with *E. keratini* EPI-7^T^ was less than 50% in all strains (1110 to 1646). However, *Cutibacterium acnes* (244) and *Micrococcus luteus* (298) had a relatively high ratio of AAI that was more than 90% and less than 50% with *E. keratini* EPI-7^T^ (Appendix A).

OrthoVenn2 was used to confirm orthologous gene clusters and the pairwise heatmap based on the amino acid sequence. The pairwise heatmap showed a correlation based on the number of overlapping clusters of *E. keratini* EPI-7^T^ and the skin flora group. As a result, *C. acnes* and *M. luteus* showed the highest associations with *E. keratini* EPI-7^T^ in amino acid identity. This result corresponds with PATRIC analysis. Although the correlation with other strains was low, the correlation between strains belonging to the same genus was close to each other (Figure 6A). Then, in GO enrichment, GO categories and GO terms of the clusters were shared by the seven comparison strains. The 311 orthologous gene clusters across all strains are involved in 266 biological process (BP), 41 molecular function (MF), and 1 cellular component (CC) GO term. Of those GO terms, “Translation” (GO:0006412, 46 clusters); “SOS response” (GO:0009432, 8 clusters); “‘de novo’ IMP biosynthetic process” (GO:0006189, 6 clusters); “Translation elongation factor activity” (GO:0003746, 4 clusters); “Translation initiation factor activity” (GO:0003743, 3 clusters) were shown to have significant associations (Appendix A). As for the closest microorganisms, *C. acnes* and *M. luteus*, we screened that 240 orthologous gene clusters between 3 species were associated with 162 BP, 58 MF, and 10 CC GO terms. In addition, the GO enrichment shared by the three strains were significantly associated with “Terpenoid biosynthetic process” (GO:0016114), including four genes: 4-hydroxy-3-methylbut-2-enyl diphosphate reductase (HMBPP reductase), HMBPP synthase, 2-C-methyl-D-erythritol 2,4-cyclodiphosphate synthase (MEcPP), and 1-deoxy-D-xylulose 5-phosphate reductoisomerase (DXP reductoisomerase) (Figure 6B). The unique gene clusters in *E. keratini* EPI-7^T^ are implicated in the function annotation and prediction. A total of 143 unique clusters are related to 61 BP, 26 MF, and 3 CC GO terms. Those GO terms were highly enriched into “Aromatic compound catabolic process” (GO:0019439); “Oxidoreductase activity, acting on paired donors, with incorporation or reduction of molecular oxygen” (GO:0016705) (Figure 6C). Cytochromes P450 (CYPs) and luciferase-like monooxygenases (LLM) class F420-dependent oxidoreductase were involved in BP: GO:0019439. Helix-turn-helix (HTH) domain-containing protein, 2Fe-2S iron-sulfur cluster binding domain-containing protein, and Biphenyl-2,3-diol 1,2-dioxygenase 3 were involved in MF: GO:0016705.

We also checked that CDSs aligned to Clusters of Orthologous Groups (COGs) families comprising 18 functional categories (Appendix A). Among them, except for function unknown, metabolism categories that existed significantly in *E. keratini* EPI-7^T^ than in other skin flora were “Energy production and conversion” (C, 237 genes); “Amino acid transport and metabolism” (E, 399 genes); “Carbohydrate transport and metabolism” (G, 210 genes); “Coenzyme transport and metabolism” (H, 147 genes); “Lipid transport and metabolism” (I, 275 genes); “Transcription” (K, 331 genes); “Replication, recombination and repair” (L, 167 genes); “Inorganic ion transport and metabolism” (P, 245 genes); “Secondary metabolites biosynthesis, transport, and catabolism” (Q, 153 genes); and “Signal transduction mechanisms” (T, 101 genes) (Figure 7).

### 3.6. Genome-Wide Prediction of the Particular Biosynthesis Pathway in Accumulation of Orotic Acid in E. keratini EPI-7^T^

It is well known that most microorganisms are auxotrophs, relying on external nutrients from host or syntrophic microorganisms for growth and survival [48,49,50]. Nevertheless, *E. keratini* EPI-7^T^ is regarded as one autotrophic microorganism with genes necessary to synthesize 20 amino acids. Representatively, asparagine (Asn), glutamine (Gln), and serine (Ser) are essential biofactors that play the essential role of pH control and nitrogen donor through continuous bacterial metabolisms [51]. As shown in Appendix A, the biosynthetic pathway-related genes of Asn, Gln, and Ser, typically consumed the most in microorganisms, were coded in the *E. keratini* EPI-7^T^ genome. Two copies of genes (gene01609 and gene01999) can convert aspartate (Asp) to Asn. Furthermore, the other two copies of the gene (gene01563 and gene01580) encode an enzyme capable of converting Glutamate (Glu) to Gln. It also has genes encoding enzymes necessary for the process of converting 3-phosphate-D-glycerate to Ser. In addition, genes and gene clusters necessary for the biosynthesis of 20 amino acids from various substrates or other amino acids are preserved in *E. keratini* EPI-7^T^.

In addition, in the previous study, it was confirmed that *E. keratini* EPI-7^T^ used uracil as a substrate to make orotic acid, an anti-aging functional substance [3]. Based on the new annotated data information through this study, we reorganized and demonstrated gene clusters associated with orotic acid biosynthesis in the *E. keratini* EPI-7^T^ genome; (gene01830 encoding aspartate carbamoyltransferase [EC 2.1.3.2], gene01831 encoding dihydroorotase [EC 3.5.2.3], gene01832 encoding carbamoyl-phosphate synthase small chain [EC 6.3.5.5], gene01833 encoding carbamoyl-phosphate synthase large chain [EC 6.3.5.5], gene01834 encoding dihydroorotate dehydrogenase (quinone) [EC 1.3.5.2]) (Figure 8A). The biosynthesis pathway of substances that can affect skin health (riboflavin [vitamin B2], chorismate [shikimate], anthranilate, phytoene) mentioned in the previous study and related genes were confirmed to be possessed by *E. keratini* EPI-7^T^. Furthermore, they had genes involved in the production of enzymes related to skin antioxidants, such as antioxidants (trehalose), superoxide dismutase (SOD), and peroxidase (catalase, glutathione peroxidase) (Appendix A).

## 4. Discussion

In this study, we conducted a comprehensive investigation based on the newly-assembled complete genome of *Epidermidibacterium keratini* EPI-7^T^ for a deeper understanding of *E. keratini* EPI-7^T^. The results of complete genome data provide that more accurate and up-to-date comparative analysis and biosynthetic pathway prediction are possible by improving the number of functional genes and reducing the number of ambiguous genes such as unknown (DUF) or hypothetical genes compared to the previous annotation data. *E. keratini* EPI-7^T^ is still the only microorganism species belonging to the genus *Epidermidibacterium*. The statement means that in order to conduct a comparative analysis with *E. keratini* EPI-7^T^, the nine species were chosen based on their phylogenetic features which were similar to those of *E. keratini* EPI-7^T^, even if they belonged to different genera. We also tried to find common or specific features between *E. keratini* EPI-7^T^ and the comparison strains through whole-genome comparisons with the most genetically matched strains. By establishing the genetic association of these strains, we allowed researchers to make more meaningful comparisons with *E. keratini* EPI-7^T^ and provided insight into the unique characteristics and taxonomic distance of these strains.

As a result, pan-genome analysis with strains closely related to *E. keratini* EPI-7^T^ confirms that the pan-genome size increased whenever the genome of the comparison strain was added and that it was an open pan-genome with 0 < B < 1. These results generally show that the pan-genome analysis, which performs a comparative analysis of different strain levels based on the same species, is in the process of environmental adaptation and evolution according to habitat [52,53]. However, our results indicate a low evolutionary correlation between *E. keratini* EPI-7^T^ and comparison strains with a close relationship. In addition, the proportion of orthologous gene clusters in the core genome is only 6.72%, which shows that the core genome size is small because they share only the genes (DNA replication and repair gene, ribosomal RNA gene, membrane transport gene, and regulatory genes) necessary for life and survival or add genomes of different genus strains [46,54]. Therefore, it is emphasized again through the pan-genome analysis that it is a novel strain that did not exist before at the genus level.

Genome-based phylogenetic analyses between *E. keratini* EPI-7^T^ and 20 human skin microbiomes revealed genetic similarities with the genera *Micrococcus*, *Cutibacterium*, and *Corynebacterium*. It also showed independent branching from the skin microbiome (especially *Staphylococcus* and *Streptococcus*), suggesting the possibility of a skin strain that could survive independently and have specific autonomous metabolic mechanisms. Therefore, to understand the hidden ability of *E. keratini* EPI-7^T^ derived from the skin, we tried to obtain clues by conducting comparative genome analysis with representative microbes inhabiting the skin. Among comparative genomic analyses with skin flora strains and various web-based tools, OrthoVenn2 and EggNOG-mapper showed significant results. In turn, the GO enrichment shared by all comparison strains of OrthoVenn2 is essential for the survival of organisms with functions related to biological processes and molecular functions in that essential genes are more conserved [55,56]. Exceptionally, *C. acnes* and *M. luteus*, which were most associated with *E. keratini* EPI-7^T^ through pairwise heatmap, share the ability of terpenoid biosynthetic processes. These terpenoids serve a variety of roles in central cellular processes, such as electron transport, photosynthesis, membrane fluidity, signaling, and cell wall formation [57,58]. Although terpenoids were generally known as metabolites of fungi or plants, it was confirmed that terpenoids of bacterial origin exist (especially *streptomycetes*). Terpene synthase is widely encoded in bacteria [59,60,61]. In addition, it is emphasized that the GO enrichment associated with oxidative stress and organic compound catabolism, which only *E. keratini* EPI-7^T^ has, can be an important ability for stress tolerance from environmental stress by rapidly catabolizing toxic substances and reactive oxygen [62,63]. Thus, it is anticipated that *E. keratini* EPI-7^T^ may have more exceptional resistance to external factors and environmental changes in the skin than other strains.

Amino acids are known as representative natural moisturizing factors (NMFs), although a precise mechanism has not been elucidated [64]. Collagen, which is widely applied in various fields as well as used for anti-inflammatory and beautifying purposes in the skin, has a characteristic of low permeability because it does not penetrate the skin layer well when injected from the outside due to its molecular size [65,66]. Therefore, instead of using a single method of making collagen in nano units, attempts are being made to induce collagen synthesis by mixing low-molecular-weight amino acids such as glycine, proline, hydroxyproline, alanine, and glutamic acid, which are the main components of collagen [67,68,69]. In addition, amino acids can be used for medical or cosmetic purposes for treating diseases and maintaining skin according to the combination of several amino acids or with other compounds as well as collagen [70,71]. Genome-wide prediction of the particular biosynthesis pathway *E. keratini* EPI-7^T^ confirmed that the *E. keratini* EPI-7^T^ genome contains genes that biosynthesize 20 amino acids. *E. keratini* EPI-7^T^ preserves many genes related to metabolism that can synthesize necessary amino acids from other substrates or amino acids in a nutrient-poor environment such as skin. Therefore, it is predicted that by supplying amino acids necessary for other microorganisms, it forms a symbiotic relationship, inhabits epidermal keratinocytes, and plays a role in supplying amino acids. Therefore, *E. keratini* EPI-7^T^ can form a syntrophic relationship and help collagen synthesis in human skin cells while providing amino acids lacking in surrounding microorganisms and directly living in human epidermal keratinocytes [72,73].

In addition to amino acids, *E. keratini* EPI-7^T^ contains genes related to metabolite biosynthesis and enzymes, such as orotic acid, riboflavin, chorismite (shikimate), anthranilate, phytoene, antioxidants, superoxide dismutase, and peroxidase, which help various anti-aging, oxidative stress, wrinkle improvements, nutritional supplements, and moisturizing [74,75,76]. Specifically, orotic acid was well known as an intermediate in the synthesis of pyrimidines [77]. Also known as vitamin B13, orotic acid is one of the few vitamins that can help prevent skin aging [78]. In a previous study by Minsu Kang, et al., it was confirmed that the content of orotic acid in the *E. keratini* EPI-7^T^ solution was increased when uracil was added as a substrate [3]. Previously, orotic acid was predicted to be biosynthesized using uracil as a substrate. Although uracil cannot be converted to orotic acid, it can ultimately be converted to uridine monophosphate (UMP) in common with orotic acid. In addition, *pyr*R encodes an mRNA-binding attenuator, and UMP/UMT bind to *pyr*R protein and negatively regulates *pyr* expression [79,80]. Therefore, although uracil addition is not directly involved in the biosynthesis of orotic acid, we predict that the PyrR is inhibited as uracil is converted to UMP and the conversion of orotic acid to UMP is inhibited, resulting in orotic acid accumulation (Figure 8B). Indeed, Christopherson RI, et al. showed that in *Escherichia coli* k12, an exogenous donation such as adenine or uracil prevented the use of aspartate molecules in the de novo biosynthesis of pyrimidine nucleotides, resulting in an increase in aspartate concentration and a significant increase in orotic acid synthesis [81]. However, the computational in silico genome analysis only is limited in demonstrating the potential capabilities of *E*. *keratini* EPI-7^T^. Therefore, we suggest the need for abundance screening of *E*. *keratini* EPI-7^T^ in different ethnic, gender, and age groups by microbial qRT-PCR quantitative analysis utilizing EPI-7^T^-specific probes for the terms of skin microbiome research.

## 5. Conclusions

In this genome-scale study of *Epidermidibacterium keratini* EPI-7^T^, we conducted a comprehensive investigation based on the newly-assembled complete genome of *E. keratini* EPI-7^T^ for a deeper understanding of *E. keratini* EPI-7^T^. The genome-wide investigations confirmed *E. keratini* EPI-7^T^, *C. acnes*, and *M. luteus* shared similar terpenoid biosynthesis, and *E. keratini* EPI-7^T^ contained many metabolite-related genes and the abilities related to environmental stress tolerance specifically developed. Additionally, the annotation-based manual inspection speculated that the gene required for vital amino acid biosynthesis is preserved in the *E. keratini* EPI-7^T^ genome, interacting with various microorganisms and skin cells and surviving in an environment where nutrients are challenging to obtain. In addition, considering the accumulation of orotic acid in a uracil-rich environmental, we expect to easily collect target metabolites from *E. keratini* EPI-7^T^, depending on the nutrients added together. In conclusion, our study will provide important clues for future in vitro studies of EPI-7^T^ function and industrial applications such as derivation of microbial-based skin improvement materials.

## Figures and Tables

**Figure 1 microorganisms-11-00870-f001:**
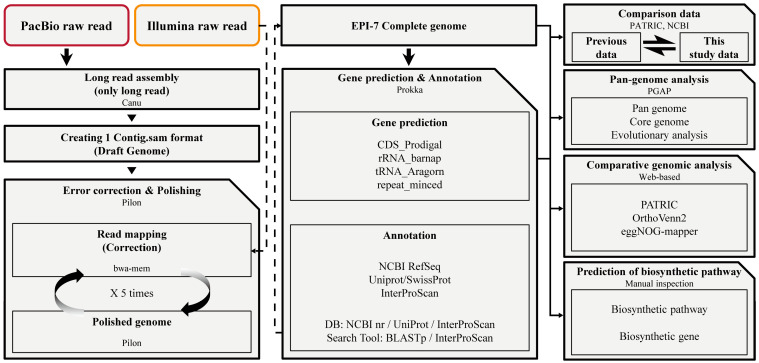
A comprehensive analysis was performed at each stage to understand *Epidermidibacterium keratini* EPI-7^T^. Step 1. Construction of new *E. keratini* EPI-7^T^ genome using PacBio raw read and Illumina raw read through hybrid de novo assembly strategy. Step 2. Gene prediction and annotation using newly-assembled *E. keratini* EPI-7^T^ genome. Step 3. Comparison between this study data and previous data. Step 4. Pan-genome analysis with strains closely related to *E. keratini* EPI-7^T^ and comparative genomic analysis with skin flora. Step 5. The discovery of functional genes and biosynthesis genes that can affect skin health.

**Figure 2 microorganisms-11-00870-f002:**
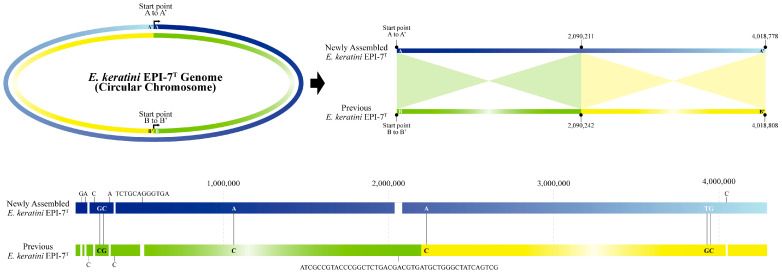
Representation of the newly-assembled and previous genome. A comparison between the newly-assembled genome and the previous genome was performed using Chromeister and the genome alignment (Mauve) services of PATRIC. In the case of the base sequence direction, when the newly-assembled genome of *E. keratini* EPI-7^T^ is expressed as a dark blue line to a light blue line (A to A′), the genome sequence of the previous *E. keratini* EPI-7^T^ is expressed as a green line to a yellow line (B to B′). It was found that the genome sequences were expressed in different orientation.

**Figure 3 microorganisms-11-00870-f003:**
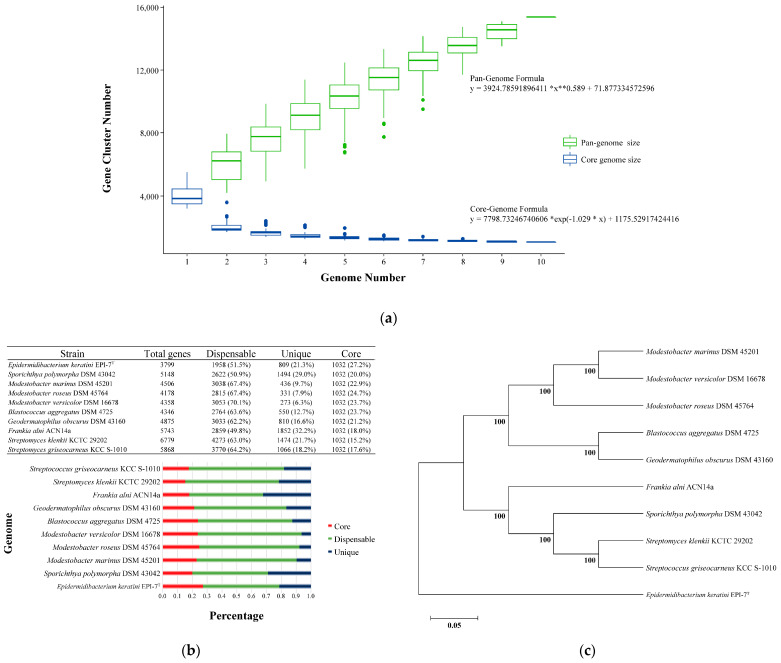
Analysis of pan-genome with species closely related to *E. keratini* EPI-7^T^. Pan-genome analysis with strains closely related to *E. keratini* EPI-7^T^ indicated that *E. keratini* EPI-7^T^ had low genetic similarity to the comparison strains, which meant that *E. keratini* EPI-7^T^ was also genetically different from the most closely-related strains. (**a**) Pan-genome size increased, and core genome decreased as the comparison strains genomes were added. (**b**) The pan-genome of 15,363 orthologous gene clusters was divided into the core-genome of 1032 orthologous gene clusters (6.72%) present in all 10 strains, the accessory genome of 5236 orthologous gene clusters (34.08%) partially present in 10 strains, and 9095 strain-specific orthologous gene clusters (59.20%). (**c**) The phylogenetic analysis based on core-genome was performed with 1000 bootstrap replications for checking an evolutionary distance.

**Figure 4 microorganisms-11-00870-f004:**
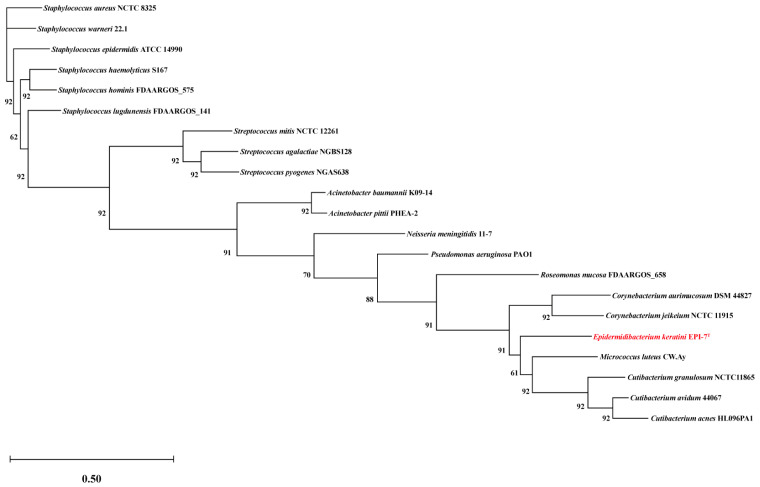
Genome-based phylogeny analysis with *E*. *keratini* EPI-7^T^ and the 20 prominent skin microbiota in humans using UBCG. Evolutionary distance based on the 92 core genes selected by UBCG shows the relevant relationship between *E*. *keratini* EPI-7^T^ and 20 human skin microbiotas. Bootstrap analysis was performed at 100 replications. Bootstrap values are given at branching points. Bar, 0.50 substitution per position. It can be confirmed that the *E*. *keratini* EPI-7^T^ (in red) is the most relevant relationship with *Micrococcus luteus*. On the contrary, the most independent branch is formed at *Staphylococcus* and *Streptococcus* genera.

**Figure 5 microorganisms-11-00870-f005:**
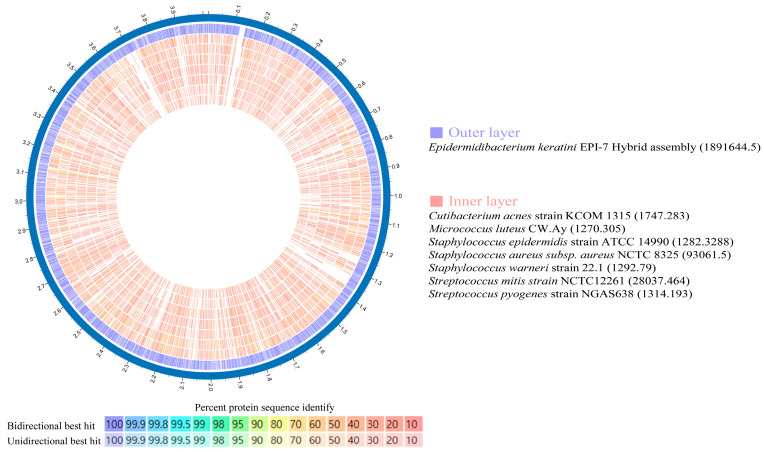
Amino acid identity (AAI) using Proteome comparison (PATRIC). Proteome comparison analysis based on amino acid sequence using PATRIC showed that *E. keratini* EPI-7^T^ had a low AAI with those of skin flora. However, *Cutibacterium acnes* (244) and *Micrococcus luteus* (298) had a relatively high AAI ratio of more than 90%, less than 50%, with *E. keratini* EPI-7^T^.

**Figure 6 microorganisms-11-00870-f006:**
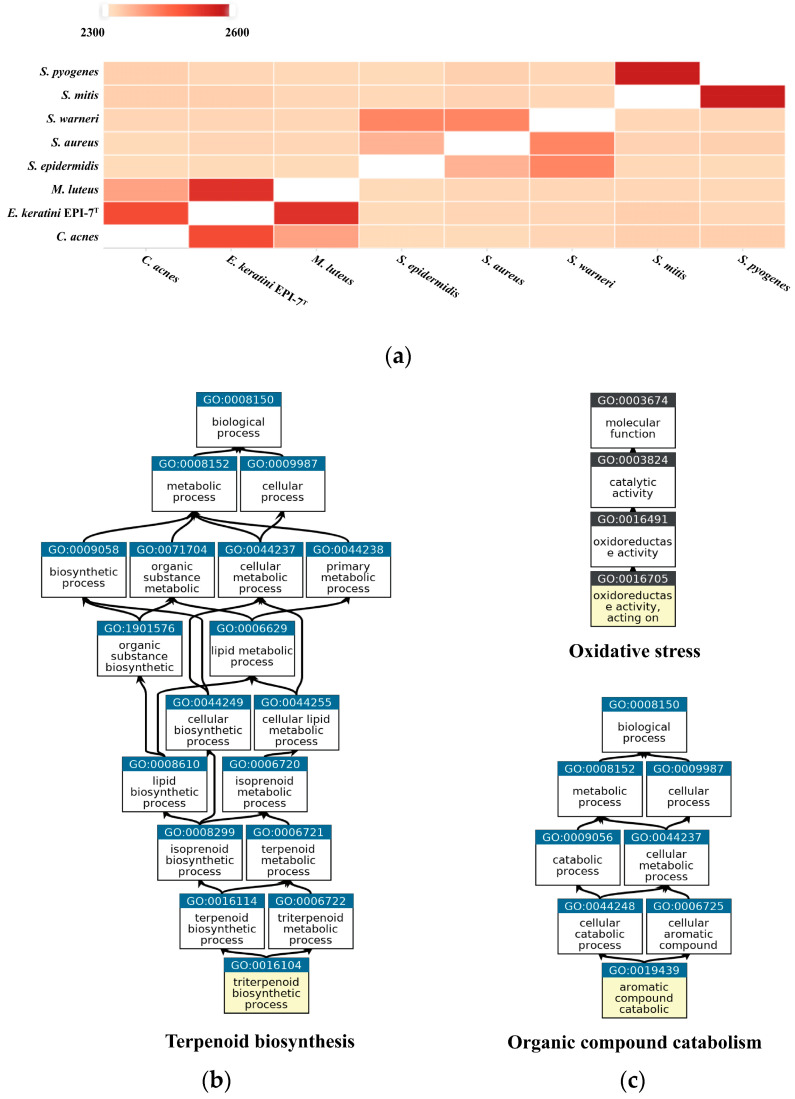
Pairwise heatmap based on amino acid sequence and GO enrichment. (**a**) In the pairwise heatmap, each cell is the overlapping cluster number between the two genomes in contact with the horizontal and vertical axes, and the darker the color, the higher the overlapping cluster number. Pairwise heatmap based on amino acid sequence confirmed that *C. acnes* and *M. luteus* showed the highest association with *E. keratini* EPI-7^T^. (**b**) GO enrichment based on GO terms showed that *C. acnes*, *M. luteus*, and *E. keratini* EPI-7^T^ shared “Terpenoid biosynthetic process”, but (**c**) “Aromatic compound catabolic process” and “Oxidoreductase activity” was unique to *E. keratini* EPI-7^T^.

**Figure 7 microorganisms-11-00870-f007:**
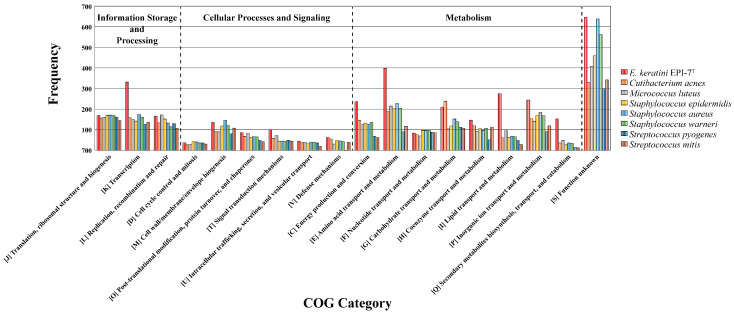
Bar chart plot showing the frequency of genes per COG category. EggNOG-mapper based on COG category frequency showed that *E. keratini* EPI-7^T^ contained significantly more genes related to the metabolism category than other strains.

**Figure 8 microorganisms-11-00870-f008:**
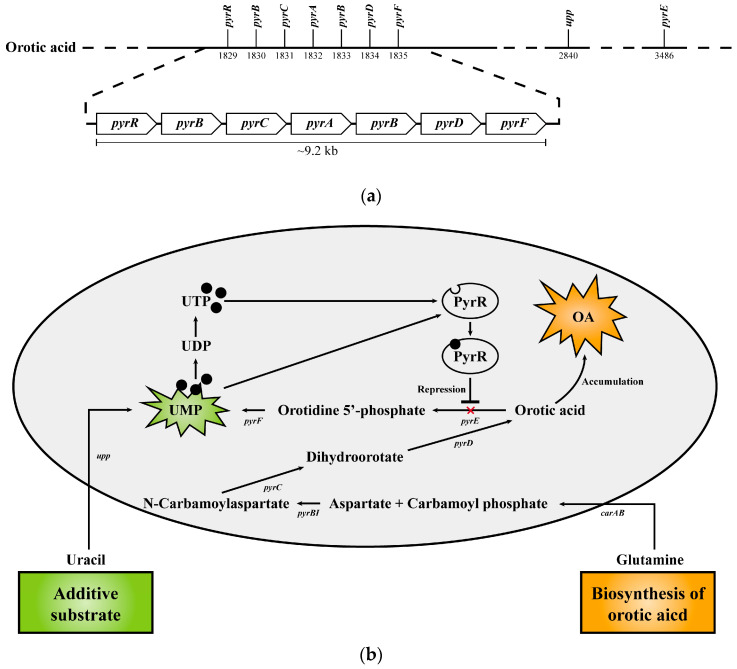
Biosynthetic pathway of orotic acid in *E. keratini* EPI-7^T^. (**a**) The gene cluster for biosynthesis of orotic acid shows the distribution in the *E. keratini* EPI-7^T^ genome. (**b**) The names of genes encoding enzymes used in each step of orotic acid biosynthesis and UMP synthesis are as follows: carAB, carbamoyl-phosphate synthase small chain/carbamoyl-phosphate synthase large chain; *pyr*BI, aspartate carbamoyltransferase catalytic subunit/aspartate carbamoyltransferase regulatory chain; *pyr*C, dihydroorotase; *pyr*D, dihydroorotate dehydrogenase; *pyr*E, orotate phophoribosyltransferase; *pyr*F, orotidine-5′-phosphate decarboxylase; *pyr*R, bifunctional *pyr* operon transcriptional regulator/uracil phosphoribosyltransferase *Pyr*R; *upp*, uracil phosphoribosyltransferase.

**Table 1 microorganisms-11-00870-t001:** Characteristics of *E. keratini* EPI-7^T^ genome.

Assembly Characteristics	Value
Number of contigs	1
Total length (bp)	4,018,778
N50 (bp)	4,018,778
N90 (bp)	4,018,778
G + C content (%)	67.34%
**Genome Characteristics**	**Value**
Genome size (bp)	4,018,778
DNA coding (bp)	3,719,702
Number of total gene	3851
Number of coding sequence gene	3799
Number of RNA genes	52
Number of CRISPR array	1
Total length of gene (bp)	3,719,702
Total length of coding sequence gene (bp)	3,707,562
RNA gene total length (bp)	8814
CRISPR total length (bp)	3326
Gene/Genome (%)	92.558

**Table 2 microorganisms-11-00870-t002:** Comparison of annotations between newly-generated and previous *E. keratini* EPI-7^T^.

Newly-Assembled *E. keratini* EPI-7^T^		Previous *E. keratini* EPI-7^T^	
Total Genes	3851	Total Genes	3851
CDS	3799	CDS	3799
rRNA	3	rRNA	3
tRNA	48	tRNA	47
tmRNA	1	tmRNA	3
Reapet Region	0	Reapet Region	0
CRISPR Array	1	CRISPR Array	1
CDSs	3799	CDSs	3799
Annotated Genes	3786	Annotated Genes	3753
Unannotated Genes	13	Unannotated Genes	46
Annotated Genes (%)	99.66%	Annotated Genes (%)	98.79%
Annotated Genes	3786	Annotated Genes	3753
Functional Genes	3169	Functional Genes	3012
Unknown (DUF)	113	Unknown (DUF)	178
Hypothetical Genes	504	Hypothetical Genes	563

## Data Availability

All data generated during this study are included in this article.

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
