# Peer review of "Genome-Wide Pathway Exploration of the Epidermidibacterium keratini EPI-7T"

_microorganisms, 2023, doi:10.3390/microorganisms11040870_

Round 1

Reviewer 1 Report

Whole-genome sequencing analysis of E. keratini EPI-7T was performed to generate new complete genome and annotation information to judge the use value of E. keratini EPI-7T. 

my remarks : 

- the summary is too over promising please correct 

- the interest is mitigate since only based on genomic analysis and annotation 

- please provide figure in HD to be able to read them  

- one or 2 vitro studies will help to realise the potential interest of this microorganisms 

Author Response

  • [Author to respond reviewer 1 - MDPI]

Reviewer 2 Report

Title: Genome-wide pathway exploration of the Epidermidibacterium keratini EPI-7T

Yunseok Oh et al. conducted a genome-wide investigation of Epidermidibacterium keratini EPI-7T. Particularly, they performed a complete genome sequencing of E. keratini EPI-7T, a comparative genomic analysis with a group of closely related skin flora strains, and an investigation of metabolic pathways involved in valuable substances with potential cosmetic functions. Overall, the goal of the work is interesting, however, as delineated in detail below, the manuscript needs some minor edits and a substantial modification of figures.

Abstract:

·       The abstract appropriately summarized the work performed in the present study, however the authors should define the acronymous “WGS” mentioned in line 21 above in the text (line 17).

Introduction and methods:

·       Line 41-42. “Through previous studies, E. keratini EPI-7T is regarded as a hidden beneficial microbe with potential applicabilities in skin anti-aging, skincare, cosmetic, human beauty industry, and skin microbiome”. The authors should add references at the end of the sentence.

·       Line 78 and 151. The authors should list the “seven strains” used in the study.

Results:

·       Figure 2. The authors should improve the quality of figures and text. A color legend might help the reader through the results.

·       Figure 3. The resolution of figures and texts needs to be improved. All bacteria names have to be reported in italic form. In panel (A) there are missing values on the x-axis.

·       Line 278. “Micrococcus luteus” has to be written in italic.

·       Figure 4. Figures and texts are unclear. The authors should improve the quality of each panel. In particular, panel (B), (C), and (D) has to be magnified. The data might be split into two figures if need it.

·       Caption of figure 4. The authors should add a title to the figure. “Cutibacterium” and “Micrococcus” have to be written with the first letter in a capital case.

·       Line 352-357. The use of alternate squared and round brackets might make the text clearer.

·       Figure 5. Text in panel (A) should be enlarged.

Discussion:

·       Line 391. The authors should list the mentioned genes. This might help the reader to follow the discussion.

·       Line 446-447. “Christopherson RI, et al.” should be written in normal format, instead “Escherichia coli k12” in italic.

·       General comments: the authors might increase the discussion about the cosmeceutical application of Epidermidibacterium keratini EPI-7T.

Patents:

·       The authors should populate this section or delete it.

Author Response

  • [Author to respond reviewer 2 - MDPI]

Reviewer 3 Report

In the manuscript “Genome-wide pathway exploration of the Epidermidibacterium keratini EPI-7T” (ID microorganisms-2020496), Yunseok Oh and colleagues performed a comprehensive study of the genome of the Epidermidibacterium keratini EPI-7T by the second and third generation sequencing technology, and the results indicate that the genome of bacteria has genes for biosynthesis of the important amino acids, orotic acid, riboflavin (B2), and chorismite. Their research provides a solid bioinformatics base for further strain development and utilization.

The main concerns:

1)    The picture of the article is very fuzzy, especially in Figure 4. Furthermore, The words on the picture can not be seen clearly.

2)    Why did the given strains be selected during the pan-genome analysis? Reasonable discussion or explanation will be appreciated for better understanding.

3)    Similarly, why did the given strains be selected for comparative genomic analysis? Whether these strains can represent the skin microbiota, and whether the comparison can be expanded to obtain more information. There are many skin microbiota data published already, it will be appreciated to analysis EP1-7T genome under skin microbiota background.

Minor points:

1)    The names of bacteria are not written properly, such as “Micrococcus luteus” in line 278 on page 8 without italics, the genus initials of “bacterium acnes and bicrococcus luteus” in lines 324 on page 10 are not capitalized.

2)     Some sentences from line 372 on page 11 to line 375 on page 12 are redundant, which are “comparison between the newly assembled genome and the previous genome was performed using Chromeister and the genome alignment (Mauve) services of PATRIC.” and “In the case of sequence orientation, when the newly assembled genome was referenced, it was found that the sequences of strands with different senses were represented.”

3)     Incomplete annotations of some figures. For example, Figure 4B does not explain the meaning of the color representation, and the annotation in Figure S2 is not particularly detailed.

Author Response

  • [Author to respond reviewer 3 - MDPI]

Round 2

Reviewer 1 Report

the revised manuscript is now good for publication / relevant improvement that bring high level of interest

Author Response

Thank you for your efforts

We added genome-based phylogeny analysis with EPI-7 and 20 species of human skin microbiota.

- [Author to respond reviewer 1 - MDPI]

Reviewer 3 Report

As I mentioned in the comments, there are a lot of skin microbiota database, it will be better for readers to understand your results, if you compare your data at the database background, but not only some selected skin strains background.

Author Response

Thank you for your suggestions

We added genome-based phylogeny analysis with EPI-7 and 20 species of human skin microbiota.

- [Author to respond reviewer 3 - MDPI]
